# Changes in DNA Methylation and mRNA Expression in Lung Tissue after Long-Term Supplementation with an Increased Dose of Cholecalciferol

**DOI:** 10.3390/ijms25010464

**Published:** 2023-12-29

**Authors:** Alicja Wierzbicka, Ewelina Semik-Gurgul, Małgorzata Świątkiewicz, Tomasz Szmatoła, Anna Steg, Maria Oczkowicz

**Affiliations:** 1Department of Animal Molecular Biology, National Research Institute of Animal Production, ul. Krakowska 1, 32-083 Balice, Poland; alicja.wierzbicka@iz.edu.pl (A.W.); ewelina.semik@iz.edu.pl (E.S.-G.); tomasz.szmatola@iz.edu.pl (T.S.); anna.steg@iz.edu.pl (A.S.); 2Department of Animal Nutrition and Feed Science, National Research Institute of Animal Production, ul. Krakowska 1, 32-083 Balice, Poland; malgorzata.swiatkiewicz@iz.edu.pl; 3Center for Experimental and Innovative Medicine, The University of Agriculture in Kraków, Rędzina 1c, 30-248 Kraków, Poland

**Keywords:** cholecalciferol, lungs, swine, mRNA, DNA methylation

## Abstract

Maintaining an appropriate concentration of vitamin D is essential for the proper functioning of the body, regardless of age. Nowadays, there are more and more indications that vitamin D supplementation at higher than standard doses may show protective and therapeutic effects. Our study identified differences in the body’s response to long-term supplementation with cholecalciferol at an increased dose. Two groups of pigs were used in the experiment. The first group received a standard dose of cholecalciferol (grower, 2000 IU/kg feed, and finisher, 1500 IU/kg feed), and the second group received an increased dose (grower, 3000 IU/kg feed, and finisher, 2500 IU/kg feed). After slaughter, lung samples were collected and used for RRBS and mRNA sequencing. Analysis of the methylation results showed that 2349 CpG sites had significantly altered methylation patterns and 1116 (47.51%) identified DMSs (Differentially Methylated Sites) were related to genes and their regulatory sites. The mRNA sequencing results showed a significant change in the expression of 195 genes. The integrated analysis identified eleven genes with DNA methylation and mRNA expression differences between the analyzed groups. The results of this study suggested that an increased vitamin D intake may be helpful for the prevention of lung cancer and pulmonary fibrosis. These actions may stem from the influence of vitamin D on the expression of genes associated with collagen production, such as *SHMT1*, *UGT1A6*, and *ITIH2*.The anti-cancer properties of vitamin D are also supported by changes in *KLHL3* and *TTPA* gene expression.

## 1. Introduction

Vitamin and mineral supplementation is becoming an increasingly integral part of both human and animal diets. Among the range of available agents, vitamin D is one of the most commonly recommended dietary supplements.

Vitamin D is unique because it can be synthesized by the body. However, skin contact with sun-emitted UV-B radiation (290–315 nm) is necessary for its synthesis. Moreover, several factors, including sex, skin color, and obesity, may influence the availability of this vitamin [1]. In humans, inadequate environmental conditions e.g., staying indoors, covering the skin, and using cosmetics with UV filters, are the main cause of vitamin D deficiencies. Similarly, in the case of animal husbandry, constant or seasonal habitation with no or only poor access to sunlight determines the need for vitamin D supplementation. Nowadays, vitamin D deficiencies are frequently diagnosed and cause multifaceted irregularities in the body. In humans, adequate vitamin D status in young individuals is one of the main elements determining their proper development and growth. Later in life, vitamin D continues to regulate the body’s calcium–phosphate balance; however, the significance of its action manifests itself evenly in other areas as well.

To date, many studies have investigated the effects of vitamin D on lung diseases of different etiologies such as pneumonia, chronic obstructive pulmonary disease, asthma, or COVID-19. However, reviews compiling this research have indicated ambiguous results [2,3,4]. A possible reason for these inconsistencies is the variety of doses and methods of vitamin D supplementation used in the experiments.

In farm animals, especially in pigs, lung diseases caused mainly by mycoplasma pneumonia infections and high ammonium concentrations in the air significantly impact the animals’ welfare and reduce breeders’ incomes. However, the maximal dose of vitamin D in pig nutrition is 2000 IU/kg, as recommended by the European Union (https://eur-lex.europa.eu/legal-content/EN/TXT/PDF/?uri=CELEX:32019R0849&rid=19, accessed on 1 May 2019). Therefore, similarly, in human nutrition, the question arises as to whether a higher dose should be used.

While the beneficial effects of vitamin D on the skeletal system are widely recognized, its effects on non-skeletal diseases are the subject to debate in the scientific community. Therefore, studies on changes caused by vitamin D supplementation at the molecular level, such as gene expression, proteins, and DNA methylation, can be important voices in this debate. Nevertheless, so far, there have been few studies describing the effect of vitamin D supplementation on gene expression in lungs on the scale of the whole transcriptome in vivo. It has been shown previously that maternal vitamin D deficiencies induce changes in 2233 transcripts in newborn rats’ lungs [5]. However, to our knowledge, the impact of the use of increased doses of vitamin D on the lung transcriptome or methylome in any model animal has not yet been investigated.

In recent years, the domestic pig has been increasingly used as a model animal in transcriptomic and epigenomic studies. Using the pig as a model may allow us to understand the changes at the molecular level that occur under the influence of vitamin D and assess its potential benefits on lung health. At the same time, this approach may contribute to the development of new recommendations in animal nutrition, contributing to improved health and well-being and reducing the amount of antibiotics used in animals on commercial farms.

The methodology used in this study enabled a bilateral analysis of supplementation-induced changes at both the epigenome and transcriptome levels. Our study—by combining changes in DNA methylation and gene expression in lung tissues—provides comprehensive knowledge on the effect of the increased long-term daily intake of chole-calciferol in healthy animals.

## 2. Results

### 2.1. Plasma Vitamin D Concentration

The average plasma 25(OH)D concentration in group 1 amounted to 39.67, and it was 63.96 ng/mL in group 2.

### 2.2. Methyl-Seq

The sequencing of RRBS libraries proceeded correctly for all 16 samples. The sequencing results are available in the Gene Expression Omnibus (GEO) database under access number GSE248607. The average number of raw reads per sample was 30.2 mln, and the average number of reads after filtering was 29.7 mln. The average number of uniquely mapped reads was 19.02 mln, which was an average of 64.16% for all the mapped reads. Detailed information on the individual samples is provided in Appendix A.

A total of 2349 CpG sides with statistically significant differences in methylation levels were identified. Of the DMSs (differentially methylated sites), 955 were hypomethylated (40.66%) and the remaining 1394 (59.34%) were hypermethylated. Most DMSs were identified on chromosomes 6 (*n* = 248) and 3 (*n* = 209). The smallest number of alterations were on the sex chromosomes (X, 45 and Y, 12) and chromosome 16 (*n* = 47). On all other chromosomes, the number of DMSs ranged from 70 to 158. A total of 1116 (47.51%) of the identified DMSs were related to genes and their regulatory sites (Appendix A), and 628 of these DMS were hypermethylated (56.27%) while 487 were hypomethylated (43.73%). Annotation of the DMSs according to the gene features revealed that most of the DMSs were in introns (47.17%) (*n* = 1108) and intergenic regions (34.40%) (*n* = 808). In contrast, the DMSs in the coding parts of the genes accounted for 5.24% (*n* = 123).

### 2.3. mRNA-Seq

Sequencing proceeded correctly for 9 of the 10 samples. One sample was rejected due to a significantly lower number of reads. The sequencing results are available in the GEO database under access number GSE242293. The average number of raw reads for the nine mRNA libraries was 11.3 million, with an average of 10 million uniquely mapped reads, constituting an average mapping rate of 88.3% for all the reads. Further details on the individual samples can be found in Appendix A.

A comparison of the transcriptomic profiles of the animals receiving the increased dose and those receiving the standard dose of cholecalciferol showed that the expression of 195 genes was significantly altered (q-value of <0.05) (Appendix A). Among the altered genes, 168 were downregulated (86.15%) while the remaining 27 were upregulated (13.85%) in the animals receiving increased doses of cholecalciferol (Appendix A). The most highly altered genes (log2FoldChange > 3 or <−3) included *EXTL1* (log2FoldChange = −4.04), *ENSSSCG00000057577* (log2FoldChange = −3.619), *ENSSSCG00000018197* (log2FoldChange = −3423), *GLP2R* (log2FoldChange = −3.36), *NPC1L1* (log2FoldChange = −3.279), *ENSSSCG00000042623* (log2FoldChange = −3.051), *SCEL* (log2FoldChange = 3.448), and *SCPEP1* (log2FoldChange = −3.091). The remaining genes all showed log2Foldchange > 0.928 or <−1.02.

### 2.4. Integration of the Methyl-Seq and mRNA-Seq Results

A comparison of the sets of genes whose expressions and methylation profiles were significantly altered under the influence of the increasing cholecalciferol dosages identified 11 genes (Table 1). However, only the changes in the methylation of the *ITIH2* gene (log2FoldChange = −2.164) were localized in the gene promoter.

### 2.5. qPCR Validation

The *ITIH2* gene, which exhibited a significant downregulation in expression and demonstrated hypermethylation in its promoter region, underwent qPCR analysis. This analysis confirmed the significant downregulation of the *ITIH2* gene detected in the mRNA and RRBS sequencing results. Additionally, we selected three other genes (*TTPA*, *UGT1A6,* and *KLKL3*) whose expression as well as methylation levels changed under the influence of the increasing cholecalciferol dosages (Figure 1). The qPCR results confirmed significant reductions in the expression of *TTPA* (*p*-value = 0.001) and *UGT1A6* (*p*-value = 0.038) genes. However, the decrease in the *KLHL3* gene expression according to the qPCR results was not statistically significant (*p*-value = 0.076).

### 2.6. Functional Analysis

#### 2.6.1. Methyl-Seq

Table 2 shows the top 10 (FDR < 0.0017 and strength > 0.3) functional effects identified by the RRBS sequencing. The analysis of a set of genes with altered methylation levels showed changes in 82 biological processes. According to the FDR values (<0.00075) and strength level (0.35), the actin filament-based process and actin cytoskeleton organization were most altered in this group. Potential effects on 39 biological functions were also observed. Within this group, according to the FDR values (<0.00031) and strength level (>0.35), the greatest effects of the increasing vitamin D intake were observed within the guanyl-nucleotide exchange factor activity, GTPase regulator activity, and actin binding. On the other hand, within the biological components, the results of the functional analysis showed effects on 26 areas. According to the FDR value (<0.00072) and strength level (>0.35), the greatest changes occurred in the actin cytoskeletons. All the results of the gene-set analysis are presented in Appendix A.

The separate analyses of the hypomethylated and hypermethylated DMSs were additionally carried out. The results of the analyses showed that hypomethylated DMSs enriched 16 biological process, 7 molecular functions, and 8 cellular components. According to the FDR and strength levels, the increasing vitamin D doses enriched the actin-binding (FDR = 0.016 and strength = 0.45) and cytoskeletal protein-binding (FDR < 0.001 and strength = 0.41) functions most potently. An analysis of the hypermethylated DMSs showed that 24 biological processes, 23 molecular functions, and 16 cellular components were enriched. According to the FDR and strength levels, the GTPase regulator activity was the most enriched (FDR < 0.001 and strength = 0.46). Interestingly, the analysis of the hypomethylated DMSs also showed a strong strength level of this molecular function (0.43; FDR = 0.02). On the other hand, according to the KEGG database, based on a set of hypermethylated DMSs, the phospholipase D signalling pathway was altered (FDR = 0.0153 and strength = 0.61).

The results of the hypomethylated/hypermethylated DMSs analysis are presented in Appendix A.

#### 2.6.2. mRNA-Seq

Table 3 shows the top 10 strongest functional changes (5 from the GO database and 5 from KEEG) caused by the increasing cholecalciferol doses based on the mRNA sequencing results. The analysis showed that the applied changes in the diets of the pigs could significantly affect 103 biological processes. Based on the FDR (<0.0001) and strength (>1.2) values, we highlighted such processes as the negative regulation of blood coagulation, regulation of blood coagulation, retinoid metabolic process, and regulation of wound-healing. In turn, among the 19 significantly altered biological functions, we identified (FDR < 0.0001 and strength > 0.2) oxidoreductase activity and catalytic activity. Moreover, eight significant differences in the biological components were identified. In this area, the largest changes (FDR < 0.001 and strength > 0.49) occurred in the extracellular region and extracellular space. The results from the KEGG database indicated the effects of the increased vitamin D intake on 24 pathways, of which as many as 12 pathways with strength values of >1 and FDR values of <0.0001 could be distinguished. The results of the gene set analysis of the mRNA-seq results are presented in Appendix A.

The results of the analysis of the set of downregulated genes showed the enrichment of 106 biological processes, 28 molecular functions, and 8 cellular components. According to the FDR and strength levels, the most significantly enriched appeared to be the propionate metabolic process (FDR = 0.0007 and strength = 2.22). According to the KEGG database, the set of downregulated genes was most strongly involved (FDR < 0.001 and strength > 1.6) in the metabolism of the xenobiotics by the cytochrome P450 and chemical carcinogenesis pathways. The set of upregulated genes was not significantly involved in any pathway, nor did they significantly enrich any of the processes. The entire results of the analysis are presented in Appendix A.

#### 2.6.3. Integration of the Methyl-Seq and mRNA-Seq Results

The analysis of a set of 11 genes, presented in Table 1, showed the significant affectation of 3 pathways (Table 4). The effects on the steroid hormone biosynthesis and retinol metabolism pathways were conditioned by the altered expression of *CYP3A22* and *HSD17B6*. The glycine, serine, and threonine metabolism pathways, on the other hand, were found to be significantly altered due to the effects on the *SHMT1* and *BHMT* genes.

## 3. Discussion

The present study showed that increasing doses of cholecalciferol caused significant changes in the transcriptomes and methylomes of swine lung tissues. We found that increased vitamin D intake significantly altered the expression of nearly 200 genes. Moreover, we identified 1116 differentially methylated sites related to genes and their regulatory sites. Among all the DMSs, only 11 were associated with genes with significant changes in expression levels (Table 1). A presumed reason for the poor correlation of the results (RRBS and mRNA) was the different number of samples used in each NGS study. However, based on the results of other researchers, it should be noted that relatively low correlations are common, especially in experiments where the effect of the studied factor is not exceptionally strong [6].

It has been assumed that, via the VDR, vitamin D regulates key mechanisms such as metabolism and cell proliferation [7,8]. Increased vitamin D intake causes activation of the VDR, which, in turn, intensifies gene transcription. However, our results indicated that more than 86% of the genes altered by the increased vitamin D intake were downregulated. Also, among the genes identified as those with altered methylation, there was little bias toward hypermethylation (56%).

A functional analysis of the methylome sequencing results showed that the increasing vitamin D intake affected the GTPase-, cytoskeleton-, and actin-related processes most significantly (Table 2). Interestingly, a recent finding showed that reduced SARS-CoV-2 lung disease severity was associated with methylation changes within processes related to GTPase and actin [9]. Moreover, it appeared that increasing the dose of cholecalciferol induced changes in the methylome associated with pulmonary fibrosis. Our results indicated methylation changes in the *MeCP2* (methyl-CpG binding protein 2) gene, which is considered a key regulator of fibrosis (Appendix A) [10]. It has been shown, for example, that *MeCP2* KO mice are resistant to pulmonary fibrosis [11]. Another study found that vitamin D could inhibit the TGFβ1 stimulation of α-smooth-muscle actin expression and polymerization and prevent the upregulation of fibronectin and collagen in fibroblasts in vitro. These results indicated that vitamin D may inhibit the pro-fibrotic phenotype of lung fibroblasts and epithelial cells [12]. Additionally, we identified four genes encoding collagen (*COL4A1*, *COL5A1*, *COL6A3,* and *COL24A1*) whose methylation levels were also changed (Appendix A). One of these, *COL4A1* is considered a target of miR-29 in pulmonary fibrosis [10].

The functional analysis of the mRNA-seq results (Appendix A) identified genes related to complement and coagulation cascades (FDR < 0.0001) and chemical carcinogenesis (FDR < 0.0001). We detected the significant downregulation of the expression of genes encoding acute phase proteins (e.g., *FBG*, *FGA*, and *FGG*) and tumor-associated genes such as *SERPINC1* and *F2 (*Appendix A). *SERPINC1* is recognized as a key gene in the processes of cancer cell proliferation and migration. It has been shown that reducing *SERPINC1* expression can be an effective treatment for lung cancer [13,14]. Furthermore, the *F2*-thrombin factor 2 gene may not only affect interactions between viral proteins and cytokine receptors [15] but also play an essential role in blood coagulation, angiogenesis, tissue repair, and vascular integrity.

Our study also showed that increasing vitamin D intake results in changes in the methylation and expression levels of genes linked to retinol and retinoid metabolism (Table 3 and Table 4). This was in line with the fact that the active form of vitamin D, combined with vitamin D receptor (VDR), forms a heterodimer with the retinoid X receptor (RXR). The heterodimer formed in this way can interact with the vitamin D gene’s response elements (VDRE) [16]. Interestingly, other researchers have shown that vitamin D supplementation can affect methylation within the RXR promoter [17].

The functional analysis of the results shared by the methylome and mRNA sequencing indicated the anti-tumor and anti-fibrotic effects of increasing the cholecalciferol dosage. We identified changes in the expression of the *PSAT1* and *SHMT1* genes, as well as methylation changes in the *SHMT1* gene. The in vitro and in vivo studies on mice by Zhu et al. showed that *PSAT1* is a strong promoter of pulmonary fibrosis and that VDR regulates the expression of this gene [18]. Moreover, an in vitro study on human cell lines showed that silencing the *PSAT1* gene resulted in the inhibition of tumor proliferation and growth in non-small cell lung cancer [19]. The *PSAT1* plays an important role in connecting pathways involving glycolysis and the biosynthesis of amino acids. Its decreased expression inhibits the synthesis of serine and, consequently, glycine. Glycine is a major component of collagen, which is the building material of connective tissue. Alterations in collagen production cause lung fibrosis and may regulate cell proliferation in lung tumors. Increasing the dose of cholecalciferol resulted in a significant decrease in *PSAT1* expression and an effect on both the glycolysis and amino acid synthesis pathways, including glycine and serine. Therefore, our findings supported the suggestions that vitamin D may be a therapeutic agent in patients with pulmonary fibrosis and lung cancer, though not only through the regulation of *PSAT1* but also the *SHMT1* gene. *SHMT1* encodes serine hydroxymethyltransferase 1, an enzyme essential for converting serine to glycine [18]. We identified the downregulation and hypermethylation of the *SHMT1* gene under the influence of increasing cholecalciferol doses. Based on these results, it could be assumed that the effect of vitamin D in inhibiting collagen production in the lungs was bidirectional: first, through the inhibition of the synthesis of serine, the material necessary for the formation of collagen, and second, through the downregulation of the enzyme directly involved in collagen synthesis. Interestingly, the *UGT1A6* gene, whose variations in expression and methylation were observed, is also significantly associated with the development of pulmonary fibrosis. There have been some indications that this gene is significantly upregulated in idiopathic pulmonary fibrosis patients compared to healthy controls [20]. Moreover, the increased expression of *UGT1A6* is also characteristic of patients with cancers, including lung cancers [21,22]. UDP-glucuronosyltransferases are a group of enzymes associated with the catabolism of drugs and xenobiotics. Therefore, it is believed that altering the expression of *UGT1A* genes, including *UGT1A6*, can significantly modulate the response to the treatment, development, and progression of cancer [22]. A decrease in *UGT1A6* gene expression under vitamin D (calcitriol) supplementation has already been observed by other researchers, though in rat liver tissues [23].

The effects of vitamin D on processes associated with pulmonary fibrosis and cancer were also suggested by changes within the *ITIH2* (*SHAP*) gene. The *ITIH2* gene in our study was the only one to have significantly altered mRNA expression via a change in methylation within the promoter. The *ITIH2* gene, through the inter-alpha-trypsin inhibitor protein it encodes, is related to the serine inhibitor group. This gene may be involved in the control of inflammation and immune processes [24]. There have been indications that the ability to regulate angiogenesis may link *ITIH2* to the process of pulmonary fibrosis [25]. A study by Garantziotis et al. showed that induced lung injury resulted in a greater than sixfold increase in *ITIH2* expression in the liver and a decrease in its expression in the lungs [25]. Similarly, in lung cancers, *ITIH2* shows significant downregulation in the altered tissues [24]. ITIH2 is a part of IaI protein, which is synthesized in the liver. IaI consists of a light chain and two heavy chains (ITIH1 and ITIH2). It is generally considered that IaI is a systemic factor that enhances angiogenesis. Histological studies have shown the strong colocalization of inter-α-trypsin inhibitor protein (IaI) in disease focus in patients with pulmonary fibrosis. In contrast, the lungs of healthy subjects were weakly stained for IaI [25]. It is noteworthy that angiogenesis in patients with pulmonary fibrosis and cancer may have a dual effect. In the early stages of the disease, it may promote tissue regeneration; however, in the exacerbated stages, it may accelerate the development of the disease. The results cited above have indicated that IaI shows higher levels in patients with fibrosis; however, the expression of the lung *ITIH2* gene decreases. This result may suggest another role, unrelated to the IaI, for *ITIH2* in lung tissues.

*KLHL3* (kelch-like family member 3) is another gene that was downregulated by the increasing cholecalciferol intake in our study. There are many indications that the *KLHL3* gene is linked to cancer development. This gene appears to be downregulated in the plasma samples of patients with lung cancers [26]. However, findings regarding the expression and regulation of this gene in lung tissues are lacking. Studies presenting the expression of this gene in tumor tissues have indicated that it can be both overexpressed and downregulated under the influence of disease [27]. Nevertheless, the role of KLHL3 in the pathogenesis of cancers formed through viral interaction appears to be well-proven [28]. Indeed, it appears that KLHL3 protein expression is significantly higher in cells expressing vIRF1 (viral interferon regulatory factor 1), as well as in cells infected with KSHV (Kaposi’s sarcoma-associated herpesvirus). Researchers have found that KLHL3 mediates the infection and replication of KSHV-induced tumorigenesis [28].

Another interesting result was the significant downregulation of the *TTPA* (alpha tocopherol transfer protein) gene, which we confirmed by the mRNA-seq (log2FoldChange = −2.297) and qPCR results. In addition, we observed significant methylation changes in the introns of this gene. TTPA binds the biologically active form of vitamin E (α-tocopherol) and plays an important role in regulating the levels of this vitamin in the body. Due to its strong antioxidant properties, numerous studies have been conducted to determine whether vitamin E supplementation can protect against cancer. The first clinical trial conducted on a large group of male tobacco smokers surprisingly showed that mortality was 2% higher in the group of those taking a daily dose of alpha-tocopherol compared to the group of those not supplemented with alpha-tocopherol [29]. The lack of protective effects of vitamin E supplementation in cancer and cardiovascular disease was also confirmed in another clinical trial conducted on a group of healthy women [30]. The change in the expression level of the *TTPA* gene we observed under the influence of vitamin D indicated a possible interaction between simultaneous dietary supplementation with vitamin D and E, and we recommend further studies in this area.

Altogether, our findings, along with those of other researchers, have suggested that increasing the intake of cholecalciferol in the daily diet may exhibit anti-cancer and anti-fibrotic effects (Figure 2). It is assumed that respiratory infections can cause chronic fibrotic reactions even up to several months after infection [31]. Therefore, our results appear to be important both for breeders of animals at risk of lung disease and from a human health point of view. It is important to highlight that our mRNA sequencing analysis was conducted on a limited number of samples exclusively from males. Nevertheless, the robustness of these findings was validated across a more extensive cohort encompassing both males and females through RRBS and qPCR testing. Nonetheless, further investigations are warranted to comprehensively explore the correlation between vitamin D and the development of pulmonary fibrosis and cancer formation.

## 4. Materials and Methods

### 4.1. Animal, Diets, and 25(OH)D Blood Serum Concentration Measurements

All the procedures conducted on live animals had the approval of the local Ethical Committee for Experiments with Animals in Cracow (Resolution No. 427/2020 dated 22 July 2020). The animals were kept under the same conditions, in individual pens, at the Research Station of the State Research Institute of Animal Production in Grodziec Śląski. In the experiment, animals from two groups were utilized. Animals in the first group (group 1) received a standard dose, while those in the second group (group 2) received increased doses of cholecalciferol (Figure 3). Both groups contained 10 individuals, with 5 males and 5 females in each group. The animals received the same feed, though it differed in the levels of cholecalciferol. The feed covered all their current requirements (grower, 30–60 kg: metabolizable energy, 13.3 MJ and total protein, 172 g/kg; finisher, 60–110 kg: metabolizable energy, 13.3 MJ and total protein, 156 g/kg).

The feeding experiment ended when the animals reached individual weights of 110 kg (88 days). Immediately after the animals were slaughtered, samples from the middle parts of the upper lobes of the left lungs were collected. The samples were stored in a freezer (−85 °C) until further analysis.

The assays of total 25(OH)D concentrations in the animal plasma samples were carried out by the ANCHEM Laboratorium from Katowice in Poland, in line with the RIA method, using a DIAsource 25OH Vitamin D total-RIA-CT Kit (Rue de Bosquet 2, 1348 Louvain-La-Neuve, Belgium) and Multigamma 1260 multidetector instrument (LKB Wallac, Turku, Finland). The plasma vitamin D levels were measured using blood samples collected at slaughter.

### 4.2. DNA and RNA Isolations, Library Construction, and Sequencing

The total DNA from 16 lung samples (8 females and 8 males) was isolated using a Wizard^®^ Genomic DNA Purification Kit (Promega, Madison, WI, USA). The concentrations of the genetic materials (DNA) were confirmed using NanoDrop™ 2000/2000c spectrophotometers (Thermo Scientific™, Waltham, MA, USA). Then, high-quality DNA samples were used to prepare the libraries. The libraries for the methyl-seq were prepared using an Ovation^®^ RRBS Methyl-Seq System 1–16 kit (Tecan, San Jose, CA, USA). The RRBS libraries were sequenced in the United States by Medical Research Foundation NGS Core using Illumina NovaSeq 6000 device (Illumina, San Diego, CA, USA) as 150 bp-paired end reads and using PhiX control.

RNA was isolated from 10 lung samples (5 samples from group 1 and 5 samples from group 2, only females) using a PureLink™ RNA Mini Kit (Invitrogen, Waltham, MA, USA). The isolated genetic materials were quantitatively and qualitatively assessed using Tapestation 2200 (Agilent, Santa Clara, CA, USA). The high-quality RNA samples were used for the library preparation using a QuantSeq 3′mRNA-Seq Library Prep Kit FWD for Illumina (Lexogen, Vienna, Austria). The procedures were carried out in accordance with the manufacturers’ recommendations. Quantitative evaluation of the prepared libraries was performed using Qubit (Thermo Scientific™, Waltham, MA, USA), while a qualitative evaluation was assessed using Tapestation 2200 device (Agilent, Santa Clara, CA, USA). Sequencing of the mRNA pooled libraries (75 bp single read) was performed using a Nextseq 5500 device (Illumina, San Diego, CA, USA). The libraries were prepared for sequencing according to the standard normalization method from the NextSeq 500 and NextSeq 550 Sequencing Systems-Denature and Dilute Libraries Guide protocol. We used a 2 nM starting library concentration and a 10% PhiX addition.

### 4.3. qPCR Validation

Twenty samples were used for the qPCR analysis. The experiment began with RNA isolation from the 10 remaining samples. RNA from 20 lung samples was reverse transcribed. We performed qPCR on the *KLHL3*, *TTPA*, *UGT1A6,* and *ITIH2* genes using *RPS29* as an endogenous control. The genes were selected based on the results of the RRBS and mRNA sequencing, and 500 ng of RNA was reverse-transcribed to cDNA using a High-Capacity RNA-to-cDNA™ Kit (Applied Biosystems™, Waltham, MA, USA). The real-time PCR was performed using TaqMan™ Fast Advanced Master Mix for qPCR (Applied Biosystems™, Waltham, MA, USA) and TaqMan Real-Time PCR assays on a QuantStudio ™ 7 Flex Real-Time PCR System (Applied Biosystems™, Waltham, MA, USA). The relative quantity data were analyzed on a Thermo Fisher Cloud (Thermo Scientific™, Waltham, MA, USA). Analysis of the results was carried out using SAS 9.4 software (SAS Institute Inc., Cary, NC, USA).

### 4.4. Statistical Analysis

#### 4.4.1. Methyl-Seq

The first step of the data analysis was the quality control of the raw sequencing reads using FastQC v. 0.12.1 software. Low-quality reads (quality level of <20 and read length of <36) and fragments containing adapter sequences were filtered using FlexBar v. 3.5.0 software. Matching to the swine reference genome (Sscrofa11.1) was performed using bisulfite mapping software-BSMAP v. 2.9.0, with the default options recommended for RRBS data specifying the enzyme cleavage site (MspI) and mapping to the two forward strands. The Methylation Caller software provided in the BSMAP v. 2.9.0 package was used to determine the percentage of methylation in the individual CpG sites with the coverage of more than 5 reads. The CpG methylation analysis included the distribution in the swine chromosomes and the distribution in the upstream regions, 5′-UTRs, 3′-UTRs, exons, introns, and intergenic regions. Next, the files were processed using R package (version 4.3) to obtain the input data for the Methylkit software. Methylkit v. 1.26.0 software was used to identify the differentially methylated sites (DMS) with cutoff values of at least 25% methylation differences between the two groups and q-values of <0.05. Gene annotations were obtained from the Sscrofa11.1 Ensembl GTF annotation.

#### 4.4.2. mRNA-Seq

The demultiplexed fastq files downloaded from the sequencing server were quality-checked, trimmed from reads, and mapped from reads using FastQC 11.8, FLEXBAR 3.5.0, and TopHat 2.1.1, respectively. Samtools 1.9, RSeQC, HTSeq-count 0.11.1 software, and Gtf-Ensembl annotation 96 were used to assess the mapping statistics and read counts. Then, to perform the differential expression analysis, the R program and DESeq 2 software suites were used. Differentially expressed genes were regarded as genes with q-values <0.05 (FDR, false discovery rate), Benjamini–Hochberg (BH) adjustments, and no fold-change thresholds. Only the genes that showed base means > 20 were used for further analyses. The gene annotations were obtained from the Sscrofa11.1 Ensembl GTF annotation.

#### 4.4.3. Integration of the Methyl-Seq and mRNA-Seq Results

We used the RRBS and RNA-seq datasets to relate the changes in the CpG site methylation to the changes in the expression levels of the associated genes. The Venny 2.1 program was used for this comparison.

#### 4.4.4. Functional Analysis

An analysis of the methyl-seq results, mRNA-seq results, and the combined data was performed. The upregulated/hypermethylated and downregulated/hypomethylated genes were used for joint (gene set analysis (GSA)) and separate analyses.

The functional analyses were carried out using STRING software (version 12.0), with which Gene Ontology (GO) and Kyoto Encyclopedia of Genes and Genomes (KEGG) pathway enrichment analyses were performed. In addition, the BioMart (release 110) and Venny 2.1 programs were used during these analyses. The top 10 results from the functional enrichment analysis were selected based on the FDR (false discovery rate) and strength level values.

## 5. Conclusions

Our results showed the putative mechanism by which increasing vitamin D intake may reduce the risk of lung cancer and pulmonary fibrosis in healthy individuals. These actions may have been due to the effects of vitamin D on collagen production-related genes, such as *SHMT1*, *UGT1A6*, and *ITIH2*. The anticancer properties of vitamin D are further supported by changes in the expression levels of the *KLHL3* and *TTPA* genes. Changes in *TTPA* gene expression also indicated that vitamin D can affect vitamin E action. The likely epigenetic mechanism regulating these processes is DNA methylation; however, further experimental studies are needed to confirm our hypothesis.

## Figures and Tables

**Figure 1 ijms-25-00464-f001:**
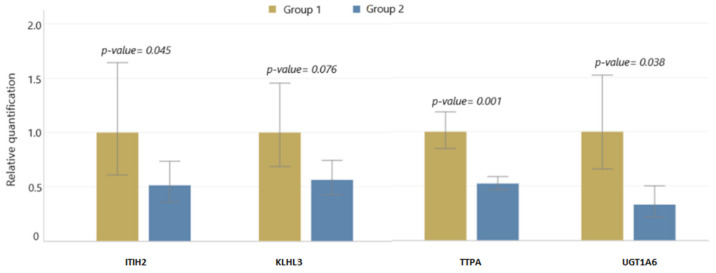
The qPCR results (RQ, relative quantification) for the groups of animals supplemented with different doses of vitamin D. The error bars represent the RQ min and RQ max values within the groups.

**Figure 2 ijms-25-00464-f002:**
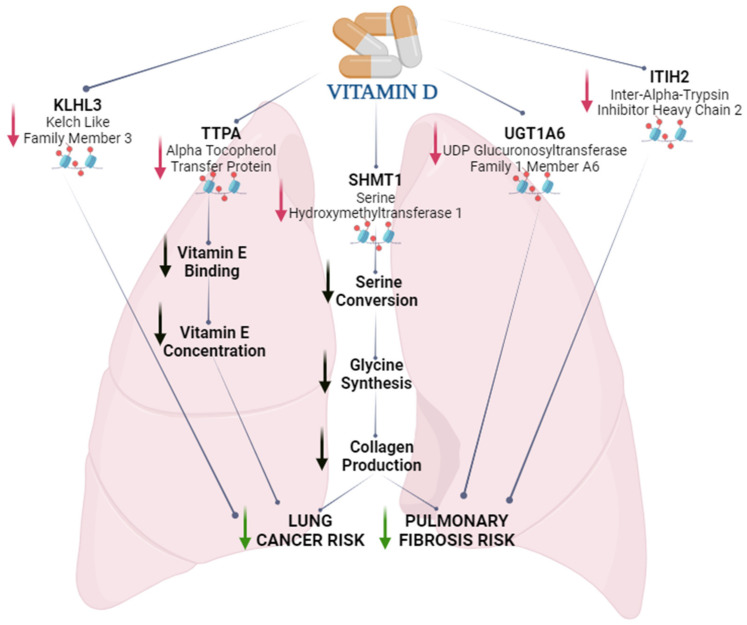
Putative molecular mechanism of the effect of increasing vitamin D intake on lung cancer and pulmonary fibrosis risks. The red and blue dots illustrate the change in gene methylation, the red arrows show decreases in gene expression, the black arrows illustrate the potential inhibition of processes, and the green arrows show the likely final effects of the changes that occurred.

**Figure 3 ijms-25-00464-f003:**
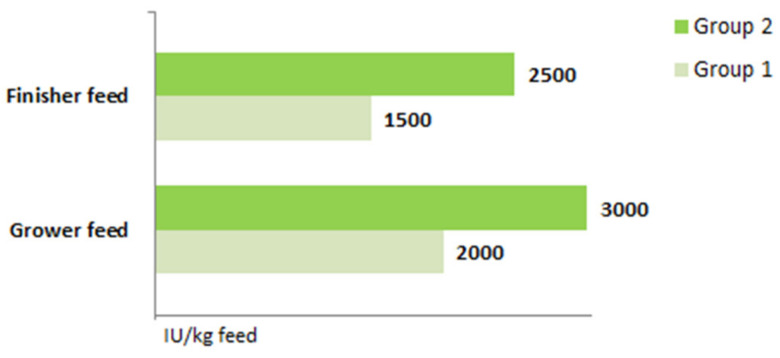
The content of cholecalciferol in the grower and finisher feeds used in the groups of animals.

**Table 1 ijms-25-00464-t001:** Similarities in the results of the mRNA-seq and methyl-seq data analyses.

Gene	mRNA-Seq		Methyl-Seq	
	Log2FoldChange	Base Mean	Meth. Diff.	Consequence
*ITIH2*	−2.164	1543.602	30.606	upstream gene
*HSD17B6*	−1.796	318.769	−32.363	intron
*CYP3A22*	−1.884	1599.78	−30.897	intron
*TTPA*	−2.297	201.53	−29.991	intron
*SHMT1*	−1.72	1007.481	−28.281	intron
*MIPEP*	−1.021	196.128	−27.29	intron
*PSMA1*	1.658	41.835	25.747	intron
*HDLBP*	−2.089	291.196	30.216	intron
*KLHL3*	−2.213	227.073	30.429	intron
*BHMT*	−2.227	1523.91	38.036	intron
*UGT1A6*	−1.603	260.41	50.914	intron

**Table 2 ijms-25-00464-t002:** Most significant results from the functional analysis of the RRBS sequencing data from the lung tissues from pigs supplemented with increased doses of cholecalciferol.

Base	Term Description	No. of Genes	Strength	FDR
GO molecular function	GTPase regulator activity	55	0.45	3.3 × 10^−7^
GO molecular function	Cytoskeletal protein binding	84	0.31	1.34 × 10^−6^
GO molecular function	Enzyme regulator activity	92	0.28	4.21 × 10^−6^
GO molecular function	Kinase binding	60	0.35	7.62 × 10^−6^
GO molecular function	Guanyl-nucleotide exchange factor activity	28	0.47	0.00025
GO molecular function	Actin binding	43	0.36	0.00031
GO biological processes	Actin filament-based process	51	0.35	0.00035
GO cellular component	Actin cytoskeleton	41	0.35	0.00072
GO biological processes	Actin cytoskeleton organization	47	0.35	0.00075

**Table 3 ijms-25-00464-t003:** Most significant results from the functional analysis of the mRNA sequencing data for the lung tissues from pigs supplemented with increased doses of cholecalciferol.

Base	Term Description	No. of Genes	Strength	FDR
KEGG	Chemical carcinogenesis	12	1.55	1.66 × 10^−12^
KEGG	Metabolism of xenobiotics by cytochrome P450	11	1.57	9.57 × 10^−12^
KEGG	Retinol metabolism	10	1.52	2.77 × 10^−10^
KEGG	Drug metabolism–cytochrome P450	9	1.5	3.59 × 10^−9^
GO biological process	Negative regulation of blood coagulation	8	1.53	2.94 × 10^−7^
GO biological process	Regulation of blood coagulation	9	1.4	2.94 × 10^−7^
GO biological process	Regulation of wound-healing	10	1.21	8.14 × 10^−7^
GO biological process	Blood coagulation	10	1.17	0.0000017
GO biological process	Retinoid metabolic process	8	1.29	7.68 × 10^−6^
KEGG	Tyrosine metabolism	5	1.34	0.00022

**Table 4 ijms-25-00464-t004:** Results of the functional analysis of the set of 11 genes identified in the RRBS and mRNA sequencing data from the lung tissues from pigs supplemented with increased doses of cholecalciferol.

Base	Term Description	No. of Genes	Strength	FDR
KEGG	Steroid hormone biosynthesis	2	2.03	0.0387
KEGG	Glycine, serine, and threonine metabolism	2	2.11	0.0387
KEGG	Retinol metabolism	2	2	0.0387

## Data Availability

The sequencing results have been deposited in the NCBI GEO database under access numbers GSE248607 (RBBS data) and GSE242293 (mRNA data).

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
