# Peer review of "Changes in DNA Methylation and mRNA Expression in Lung Tissue after Long-Term Supplementation with an Increased Dose of Cholecalciferol"

_ijms, 2023, doi:10.3390/ijms25010464_

Round 1

Reviewer 1 Report

Comments and Suggestions for Authors

Changes in DNA methylation and mRNA expression in lung tissue after long-term supplementation with an increased dose of cholecalciferol.

Wierzbicka et al conducted RRBS and RNA-seq analyses in pigs subjected to increased cholecalciferol doses, revealing that heightened vitamin D levels may mitigate inflammation in healthy individuals and hold potential for lung cancer treatment. Despite the authors' comprehensive presentation of detailed results and extensive explanations, a meticulous validation of their findings was lacking, and a suboptimal correlation was observed between RNA-seq and RRBS data. My comments below advocate for a significant manuscript revision.

1. Poor Manuscript Organization:

The manuscript lacks coherence, with descriptive results placed in the results section and conclusions confined to the discussion section. A reorganization is necessary, relocating descriptive results to the methods section and presenting the results of 'Animal, Diets, and 25(OH) D blood serum concentration measurement' in the results section.

2. Low Correlation Between RRBS and RNA-seq Data:

The authors identified 2349 DMSs using 16 samples (8 females and 8 males), 195 DEGs using 10 female lungs, with only 11 genes with integrated analysis. The poor correlation is unexplained, possibly due to unmatched samples. To address this, the authors should use the same samples for both RNA-seq and RRBS experiments, considering the variations in DNA methylation and gene expression patterns among individuals and between sexes.

3. Inadequate Functional Analyses:

3.1 When conducting GO or KEGG analyses, the authors should differentiate between hypomethylated/hypermethylated DMSs or up-regulated/down-regulated genes.

3.2 The assertion that an increased vitamin D dose counteracts inflammation requires supporting evidence. The authors should showcase phenotypical differences in lungs between treated and control groups. For instance, do treated lungs exhibit fewer signs of inflammation? Besides, Inducing inflammation in these pigs would help to validate conclusions drawn from sequencing data.

Comments on the Quality of English Language

Please verify spelling and punctuation usage.

Line 40, "defi-ciency" should be "deficiency."

Line 67, "ani-mal species" should be "animal."

Table 1: Replace commas with decimal points.

Author Response

Reviewer 1

  1. Poor Manuscript Organization:

The manuscript lacks coherence, with descriptive results placed in the results section and conclusions confined to the discussion section. A reorganization is necessary, relocating descriptive results to the methods section and presenting the results of 'Animal, Diets, and 25(OH) D blood serum concentration measurement' in the results section.

Thank you for this comment, the 25(OH)D measurement results have been moved to the correct place (Lines 82-84). Moreover, the section describing the methodology for functional analysis of NGS data has been moved to the "Materials and Methods" section (Lines 478-480).

Some descriptive results are still in discussion section because they are in supplementary tables and it is difficult to follow our reasoning without it.   (Lines 251, 261, 265)

  1. Low Correlation Between RRBS and RNA-seq Data:

The authors identified 2349 DMSs using 16 samples (8 females and 8 males), 195 DEGs using 10 female lungs, with only 11 genes with integrated analysis. The poor correlation is unexplained, possibly due to unmatched samples. To address this, the authors should use the same samples for both RNA-seq and RRBS experiments, considering the variations in DNA methylation and gene expression patterns among individuals and between sexes.

The likely reason for the poor correlation is the unequal number of samples in the groups used for the NGS study. The RRBS study conducted on a larger number of samples because DNA shows greater stability making it possible to obtain genetic material suitable for the preparation of RRBS libraries, while the quality of some RNA samples were not good enough to include them into analysis.

In addition, as can be seen in other articles, correlation also depends on the strength of the factors involved.

Lines 233- 237 give an explanation supported by the references. Thank you for the valid observation.

  1. Inadequate Functional Analyses:

3.1 When conducting GO or KEGG analyses, the authors should differentiate between hypomethylated/hypermethylated DMSs or up-regulated/down-regulated genes.

Thank you for this comment, the results of the analysis proposed by the reviewer was added (Lines 171- 184, 206-214 and Supplementary Table 7.).

The analysis proposed by the reviewer has already been carried out previously however, its results do not change the final conclusions therefore it was decided that it will not be presented.

Originally, the study reported the results of the analysis including the entire set of significantly altered genes. We opted for this method of analysis because the molecular mechanisms are intricate and while one gene is downregulated the expression of another may increase, consequently, this relationship may enrich a certain biological process or may have an effect on a specific pathway. Analyzing upregulated/hypermethylated and downregulated/hypomethylated genes separately may miss broader biological changes and functional context.

Nevertheless, the additional analysis enrich the work and may be helpful in better understanding the results.

3.2 The assertion that an increased vitamin D dose counteracts inflammation requires supporting evidence. The authors should showcase phenotypical differences in lungs between treated and control groups. For instance, do treated lungs exhibit fewer signs of inflammation? Besides, Inducing inflammation in these pigs would help to validate conclusions drawn from sequencing data.

Thank you for the pertinent comment.

Information regarding the effect of vitamin D on inflammation was excluded from the conclusions. In order to verify such a claim, it would be necessary to conduct an experiment involving animals in which inflammation was induced. In our experiment, healthy animals were used. No differences in the appearance of the lungs were observed during material collection.

The content of the "Conclusions" section has been revised. It now describes only the well-validated results of the study (Lines 487-493).

Please verify spelling and punctuation usage.

Corrected in the lines 44 and 55.

Line 40, "defi-ciency" should be "deficiency."

Corrected.

Line 67, "ani-mal species" should be "animal."

Corrected.

Table 1: Replace commas with decimal points.

Corrected.

Reviewer 2 Report

Comments and Suggestions for Authors

1. The authors are requested to provide the data of control group (without any treatment of cholecalciferol).

2. The authors are advised to provide human subject related in vitro data to strengthen their claims.

3. The authors are also requested to provide the intricate molecular mechanism(s) behind their claims.

Comments on the Quality of English Language

Moderate editing of English language required

Author Response

Reviewer 2

  1. The authors are requested to provide the data of control group (without any treatment of cholecalciferol).

Thank you for your comment. In our study, we focused on evaluating the effects of increasing the vitamin D dose compared to the standard dose, taking into account the already well-studied effects of vitamin D deficiency. The control group took the standard dose, which served as a baseline.

  1. The authors are advised to provide human subject related in vitro data to strengthen their claims.

Right point, supporting the results with this type of research can increase the credibility of the results obtained. Nevertheless, when it comes to examining the effects of vitamin D, its effect is quite well documented in in vitro studies (including the anti-inflammatory one), while there are only limited in vivo results that would confirm these observations. Therefore, it seems that our results are valuable in assessing the role of vitamin D in influencing respiratory health. In the discussion section, some references describing the results of in vitro culture of human cell lines can be found:

-19. Yang Y, Wu J, Cai J, He Z, Yuan J, Zhu X, Li Y, Li M, Guan H. PSAT1 regulates cyclin D1 degradation and sustains proliferation of non-small cell lung cancer cells. Int J Cancer. 2015 Feb 15;136(4):E39-50. doi: 10.1002/ijc.29150. epub 2014 Sep 2. PMID: 25142862.

-28. Qi X, Yan Q, Shang Y, Zhao R, Ding X, Gao SJ, Li W, Lu C. A viral interferon regulatory factor degrades RNA-binding protein hnRNP Q1 to enhance aerobic glycolysis via recruiting E3 ubiquitin ligase KLHL3 and decaying GDPD1 mRNA. Cell Death Differ. 2022 Nov;29(11):2233-2246. doi: 10.1038/s41418-022-01011-1. Epub 2022 May 10. PMID: 35538151; PMCID: PMC9613757.

In addition, the manuscript repeatedly uses the results of more popular in vitro studies that used lung or tumor cells of mouse origin:

 -13.   Ramirez AM, Wongtrakool C, Welch T, Steinmeyer A, Zügel U, Roman J. Vitamin D inhibition of pro-fibrotic effects of transforming growth factor beta1 in lung fibroblasts and epithelial cells. J Steroid Biochem Mol Biol. 2010 Feb 15;118(3):142-50. doi: 10.1016/j.jsbmb.2009.11.004. Epub 2009 Nov 17. PMID: 19931390; PMCID: PMC2821704.

- 15.   Zhang J, Tang Z, Guo X, Wang Y, Zhou Y, Cai W. Synergistic effects of nab-PTX and anti-PD-1 antibody combination against lung cancer by regulating the Pi3K/AKT pathway through the Serpinc1 gene. Front Oncol. 2022 Aug 3;12:933646. doi: 10.3389/fonc.2022.933646.

Or articles describing results from both in vitro and in vivo studies:

-30. Garantziotis S, Zudaire E, Trempus CS, Hollingsworth JW, Jiang D, Lancaster LH, Richardson E, Zhuo L, Cuttitta F, Brown KK, Noble PW, Kimata K, Schwartz DA. Serum inter-alpha-trypsin inhibitor and matrix hyaluronan promote angiogenesis in fibrotic lung injury. Am J Respir Crit Care Med. 2008 Nov 1;178(9):939-47. doi: 10.1164/rccm.200803-386OC.

-18. Zhu W, Ding Q, Wang L, Xu G, Diao Y, Qu S, Chen S, Shi Y. Vitamin D3 alleviates pulmonary fibrosis by regulating the MAPK pathway via targeting PSAT1 expression in vivo and in vitro. Int Immunopharmacol. 2021 Dec;101(Pt B):108212. doi: 10.1016/j.intimp.2021.108212

With the latter article appearing to be the most relevant because it also points to the therapeutic potential of vitamin D for pulmonary fibrosis.

Information on the type of studies cited is included in the body of the manuscript (lines 278-283).

  1. The authors are also requested to provide the intricate molecular mechanism(s) behind their claims.

The probable molecular mechanism of the anti-cancer and anti-fibrotic effects of increased vitamin D intake developed from the well-validated results of our study is presented as Figure 3.

Thank you for this comment, the added material enriches our manuscript and makes it easier to understand the presented results.

Round 2

Reviewer 1 Report

Comments and Suggestions for Authors

N/A

Reviewer 2 Report

Comments and Suggestions for Authors

The authors have tried to answer all my comments but final decision is dependent on the Editor of the journal.